# Detection of Hydroxychloroquine Retinopathy via Hyperspectral and Deep Learning through Ophthalmoscope Images

**DOI:** 10.3390/diagnostics13142373

**Published:** 2023-07-14

**Authors:** Wen-Shuang Fan, Hong-Thai Nguyen, Ching-Yu Wang, Shih-Wun Liang, Yu-Ming Tsao, Fen-Chi Lin, Hsiang-Chen Wang

**Affiliations:** 1Department of Ophthalmology, Dalin Tzu Chi Hospital, Buddhist Tzu Chi Medical Foundation, Chiayi 62247, Taiwan; wsfan@tzuchi.com.tw (W.-S.F.); s19001052@gmail.com (C.-Y.W.); 2Department of Mechanical Engineering, National Chung Cheng University, Chiayi 62102, Taiwan; nguyenhongthai194@gmail.com (H.-T.N.); start90301@gmail.com (S.-W.L.); d09420002@ccu.edu.tw (Y.-M.T.); 3Department of Ophthalmology, Kaohsiung Armed Forces General Hospital, Kaohsiung 80284, Taiwan; 4Hitspectra Intelligent Technology Co., Ltd., Kaohsiung 80661, Taiwan

**Keywords:** hydroxychloroquine retinopathy, hyperspectral imaging, artificial neural networks

## Abstract

Hydroxychloroquine, also known as quinine, is primarily utilized to manage various autoimmune diseases, such as systemic lupus erythematosus, rheumatoid arthritis, and Sjogren’s syndrome. However, this drug has side effects, including diarrhea, blurred vision, headache, skin itching, poor appetite, and gastrointestinal discomfort. Blurred vision is caused by irreversible retinal damages and can only be mitigated by reducing hydroxychloroquine dosage or discontinuing the drug under a physician’s supervision. In this study, color fundus images were utilized to identify differences in lesions caused by hydroxychloroquine. A total of 176 color fundus images were captured from a cohort of 91 participants, comprising 25 patients diagnosed with hydroxychloroquine retinopathy and 66 individuals without any retinopathy. The mean age of the participants was 75.67 ± 7.76. Following the selection of a specific region of interest within each image, hyperspectral conversion technology was employed to obtain the spectrum of the sampled image. Spectral analysis was then conducted to discern differences between normal and hydroxychloroquine-induced lesions that are imperceptible to the human eye on the color fundus images. We implemented a deep learning model to detect lesions, leveraging four artificial neural networks (ResNet50, Inception_v3, GoogLeNet, and EfficientNet). The overall accuracy of ResNet50 reached 93% for the original images (ORIs) and 96% for the hyperspectral images (HSIs). The overall accuracy of Inception_v3 was 87% for ORIs and 91% for HSI, and that of GoogLeNet was 88% for ORIs and 91% for HSIs. Finally, EfficientNet achieved an overall accuracy of 94% for ORIs and 97% for HSIs.

## 1. Introduction

The retina is situated within the inner layer of the eyeball wall and is a delicate and intricate structure. The average thickness of the human retina is approximately 250 µm [1,2]. Histologically, it consists of 10 layers extending from the retinal pigment epithelium (RPE) to the inner limiting membrane. The retina is similar to the negative film of a camera because it is responsible for photosensitive imaging. A variety of ocular diseases can arise in the retina, with some of the most common ones being age-related macular degeneration (AMD), diabetic retinopathy (DR), retinitis pigmentosa, and glaucoma [3,4,5,6,7,8,9,10].

Hydroxychloroquine (HCQ), also referred to as quinine, is primarily utilized to treat diverse autoimmune diseases, including systemic lupus erythematosus, rheumatoid arthritis, and Sjogren’s syndrome [11,12,13,14,15,16,17]. Given its strong affinity with melanin, HCQ tends to accumulate in melanin-rich tissues, such as RPE, which serves as the backbone of the retina. When utilized as part of the retinal visual cycle, HCQ can disrupt RPE metabolism, induce lysosomal damages, and ultimately lead to late irreversible retinal damages, resulting in permanent vision loss. Cell damage persists even after drug withdrawal. Nonetheless, early detection of retinopathy increases the chances of preserving vision.

The increasing popularity of artificial intelligence (AI) has led to numerous studies on using medical images to assist doctors in detecting lesions. In ophthalmology, previous research utilized ophthalmoscope images to identify diseases such as AMD, DR, and glaucoma. Lam et al. utilized AlexNet and GoogLeNet to detect DR and achieved a sensitivity of 95% [18]. Gao et al. used a deep convolutional neural network to evaluate the severity of DR in fundus images and achieved an accuracy of 88.72% [19]. Yang et al. employed ResNet50 to identify glaucomatous optic neuropathy for training and prediction, with an accuracy of 93.4% and 81.8%, respectively [20]. Li et al. also used ResNet101 to detect glaucomatous optic neuropathy by using color fundus images [21]. The results achieved an accuracy of up to 94%, a sensitivity of 96%, and a specificity of 93%. Sarki et al. reviewed deep learning research conducted from April 2014 to January 2020 that identified ocular diseases, such as DR, glaucoma, and cataract, and analyzed the image preprocessing and algorithms used [22]. Fundus camera imaging and image preprocessing steps were employed, including green channel extraction, histogram equalization, contrast enhancement, and various image transformation methods, such as rotation, resizing, and lighting correction. Algorithms used in these studies include VGG19, AlexNet, InceptionV3, ResNet50, and GoogLeNet, and their results highlight the potential of combining ophthalmoscope images and AI to aid in disease analysis.

Many scholars have utilized hyperspectral imaging technology to obtain additional information from ophthalmoscope images. Kashani et al. used hyperspectral computed tomographic spectroscopy to measure light absorbance, calculate oxygen concentration in retinal blood vessels, and analyze the degree of retinopathy in diabetic patients [23]. Hadoux et al. utilized a metabolic hyperspectral retinal camera to analyze the spectral information of the retina and create an index HS score that can effectively distinguish patients with dementia [24]. The combination of hyperspectral imaging and fundus cameras provides additional information for analysis and application. More et al. proposed a method for the early detection of Alzheimer’s disease by using retinal biomarkers from hyperspectral images [25]. HS imaging is considered a non-invasive and cost-effective method and shows promising results.

Hyperspectral conversion technology can transform images into spectra and vice versa. It is considered a “big data” method because it provides rich and useful spectral information. Deep learning models have been applied to fully exploit the potential of hyperspectral images. In a 2020 study, we combined hyperspectral images with machine learning algorithms, such as principal component analysis, to statistically analyze the classification of different stages of DR [4]. However, the utilization of hyperspectral data for the diagnosis of eye-related diseases is presently constrained. This study aims to harness the capabilities of deep learning algorithms to fully exploit the potential of hyperspectral images in comparison to ophthalmoscopic images. By employing conventional deep learning models, our objective is to reveal vital information inherent in the hyperspectral image data, thus offering valuable insights for future research in the field of ophthalmoscopic image processing.

In this study, fundus images were analyzed to detect lesions. A total of 176 fundus images were obtained from 31 patients. Hyperspectral imaging technology was used to obtain the spectrum of a specific area of the image and analyze spectral differences between lesions. Based on spectral analysis, the difference between the color fundus images of patients treated with HCQ and those of normal individuals was the most prominent in the range of 500–600 nm. Hyperspectral images (HSIs) are generated by the color reproduction algorithm through specific bands, primarily spectral bands that include wavelengths between 500 and 600 nm. Both datasets, hyperspectral images (HSIs) and original images (ORIs), were utilized for deep learning training. The ORIs were fed into HS imaging algorithms to convert them into HSIs. The HSIs and ORIs were then separately trained using four convolutional neural network models, including ResNet50 [26], Inception_v3 [27], GoogLeNet [28], and EfficientNet [29].

## 2. Materials and Methods

### 2.1. Data Collection

The overall research process is illustrated in Figure 1. In data collection, retinal images were obtained using an ophthalmoscope and cropped in a specific area. Cropping was conducted referring to the research results of Hadoux, X et al. [24]. The instruments used were Nonmyd 7 Retinal and D7200 Camera from Kowa American Corporation and Nikon Taiwan, respectively. Foveal locations (F1, F2) from [24] were merged into one area (F), as shown in Figure 2a,b. To ensure a correlation between spectral variables and avoid selection bias, regions with high color contrast representation are carefully chosen. These areas are selected based on the presence of vascular organizations and important biological structures of the eye, such as the fovea and nerve fibers. By focusing on these regions, the spectra bands are enhanced, contributing to a more effective analysis [24].

In this study, we recruited 25 patients treated with HCQ and 66 people with normal vision. According to the findings presented in Table 1, no statistically significant differences were observed between the HCQ group and the normal group in terms of age and sex. Furthermore, there were no significant differences detected between the two groups regarding the prevalence of high blood pressure, glaucoma, AMD, or DR.

We obtained 176 color fundus images, following the exclusion of blurred or affected images caused by light sources. Patients treated with HCQ were included in the study. For patients taking HCQ, the dose used was 200 mg per day, and 5 years of continuous treatment will cause retinopathy. Such a long-term continuous treatment would cause difficulty in collecting patient cases. By contrast, patients with dementia, DR, AMD, and glaucoma were excluded. The presence of dementia, DR, AMD, and glaucoma was tested. The dataset comprised 66 normal color fundus images and 110 color fundus images of patients who were administered with HCQ. Normal fundus images were defined as people without HCQ treatment, dementia, DR, AMD, glaucoma, and poor vision. A spectral-domain OCT image of the left eye of a normal individual with a normal foveal profile was used for comparison (Figure 2c). Optical coherence tomography (OCT) was used as the basis for diagnosis to accurately discern the presence of retinopathy caused by HCQ, given the difficulty in identifying HCQ on color fundus images. In Figure 2d, HCQ leads to the loss of the external membrane and ellipsoidal region and the thinning of the outer nuclear layer on both sides of the fovea (green box). The outer nuclear layer, external membrane, and ellipsoidal region remain unaffected below the fovea, forming a saucer-like appearance (red box). The fundus images collected in this study were sampled in specific areas, namely, above the temporal vascular arcade (S1, S2), the fovea (F), and below the temporal vascular arcade (I1, I2), with an area size of 240 × 240 pixels (Figure 3).

### 2.2. Data Preprocessing and Training Deep Learning Model

In data preprocessing, the image was used for data purification. Blurred images or those affected by the light source were removed, and the final data amplification included flipping, rotating, limiting contrast-adaptive histogram equalization, Gaussian blur, green channel extraction, and hyperspectral image conversion.

The hyperspectral imaging algorithm for ophthalmoscopic images is described in Appendix A and in our previous studies [4,30]. These images were then converted into 401 spectral bands spanning the visible light spectrum (380 nm to 780 nm) by using hyperspectral conversion technology. By analyzing spectral differences among normal and HCQ at different sampling positions, we aimed to identify potential biomarkers of these conditions. Based on spectral analysis, the difference between color fundus images of patients with HCQ and those of normal individuals is the most significant in the range of 500–600 nm. Therefore, spectral information within this range was selected for conversion into HSIs. Subsequently, the ORIs and obtained HSIs were subjected to testing using four artificial neural network models, including ResNet50, Inception_v3, GoogLeNet, and EfficientNet. This study aimed to compare the performance of these deep-learning models in diagnosing HCQ by using two distinct types of datasets, namely, ORIs and HSIs. A comprehensive overview of the distribution of data in the train and test sets is provided in Table 2.

The deep learning framework PyTorch was utilized for the purpose of implementing transfer learning to enhance the accuracy of diabetic retinopathy detection. Four distinct neural network models were employed in this process. The cross-entropy loss function was employed as the optimization objective, gradually decreasing after each epoch to fine-tune the model weights. A batch size of 16 was set, and the training procedure consisted of 50 epochs. The initial learning rate was configured as 0.001. Furthermore, the learning rate was reduced to 0.1 times its original value every seven epochs to aid in convergence and optimize training progress.

### 2.3. OCT System—Type B Ultrasonic Scanner (Nidek RS-3000)

The Type B ultrasonic scanner (Nidek RS-3000, NIDEK Co., Ltd., Gamagori, Japan) incorporates a choroidal mode, which offers a comprehensive assessment of the choroid, retina, and glaucoma analysis. The advanced mode of the RS-3000 allows for ultra-low sensitivity measurements, depending on the specific pathology being evaluated. With its 9 mm × 9 mm wide-area scan, it provides excellent coverage of the entire retinal structure. The unique Eye Tracer technology utilizes fundus information obtained from high-definition images to ensure precise measurements. By combining positioning, tracking, and automatic shooting functions, the Eye Tracer technology enables convenient and rapid measurements.

During macular line scans, the “Tracking HD” function compensates for micro-tracking and other involuntary eye movements. This compensation ensures that up to 120 macular scan images are aligned, enhancing image averaging. Subsequent images are precisely aligned with the baseline data, resulting in high reproducibility. The automatic registration function compensates for any adjustments made during the image acquisition process, thereby improving the quality of the subsequent data.

## 3. Results

Given that there were no significant differences observed between the HCQ and normal groups concerning age (*p* = 0.75, 95% CI: −5.51–4.86, unpaired two-tailed *t*-test, Table 1) and sex (*p* = 0.10, 95% CI: −0.02–0.05, chi-square test, Table 1), a sample size of *n* = 91 was determined to examine the spectral differences more comprehensively. Figure 4 presents the analysis of spectral differences in HCQ color fundus images between the HCQ group (*n* = 25) and the Normal group (*n* = 66) at specific sampling locations, namely (a) F, (b) I1, (c) I2, (d) S1, and (e) S2. Remarkable variations were observed in the wavelength range of 500–600 nm within the I1, I2, S1, and S2 regions. Furthermore, a minor difference was found in the HCQ spectrum between the short and long wavelength ranges. The morphological and spectral characteristics of the purified populations of melanosomes and lipofuscin granules from the human retinal pigment epithelium (RPE) changed with respect to HCQ treatment. HCQ disrupts retinal pigment epithelium metabolism, and lysosomal damage will result in retinal pigmented epithelium and their changes with melanin bleaching [30,31,32,33]. The evident disparity observed in the wavelength range of 500–600 nm may be attributed to retinopathy induced by HCQ.

This study utilized normal fundus images (i.e., those without lesions) and compared the spectra based on age groups. The reflection intensity of the spectrum between 380 and 530 nm was higher for individuals aged 80–89 years compared with those aged 60–69 and 70–79 years. By contrast, the spectral reflection intensity decreased significantly between 530 and 780 nm for individuals aged 80–89 years, but the decrease was not as prominent for those aged 70–79 years. Figure 5 provides a visualization of this phenomenon in the S1 region. One possible explanation is that as people age, the arteries and veins in the eye may shrink, and the lutein content in the eye may decrease, leading to a decrease in the reflection intensity in the long wavelength part of the spectrum. Thus, the ophthalmoscope image spectrum may exhibit different reflection intensities across different age groups.

Excluding potential analytical deviations due to diabetes, this study conducted an additional analysis of the spectrum of diabetic color fundus images. The oxygen concentration in blood vessels in the retina of patients with diabetes was primarily analyzed as the staging basis. The diabetic fundus color image spectrum was categorized into Normal, background retinopathy (BDR), preproliferative retinopathy (PDR), and proliferative retinopathy (PPDR) according to the stage. In the five regions of Figure 6, no significant difference in the spectrum was observed with diabetes, indicating that the HCQ fundus color image spectra did not vary with or without diabetes (*n* = 25, *p*-value = 0.56, chi-square test).

This study compared the four deep-learning models by using two types of data sets, namely, ORIs and HSIs, for HCQ diagnosis. The deep learning model is described in Appendix A. As shown in Table 3, the overall accuracy of the ResNet50 testing model was 93% for the ORIs and 96% for the HSIs. Similarly, the Inception_v3 model had an overall accuracy of 87% for ORI and 91% for HSI, whereas the GoogLeNet model showed an overall accuracy of 88% for ORIs and 91% for HSIs. The overall accuracy of the EfficientNet model reached 94% for ORIs and 97% for HSIs. The accuracies of ResNet50, Inception_v3, GoogLeNet, and EfficientNet increased by 3% (from 93% in ORIs to 96% in HSIs), 4% (from 87% in ORIs to 91% in HSIs), 3% (from 88% in ORIs to 91% in HSIs), and 3% (from 94% in ORIs to 97% in HSIs), respectively. These results demonstrate that the use of HSIs, which provide spectral features, can improve accuracy compared with ORIs. The increase in accuracy can range from a minimum of 3% to a maximum of 4%, depending on the neural network model used. Moreover, the learning capability of different models varied with various image data. Thus, multiple models can be utilized for comparison in deep learning applications.

However, the accuracy did not provide a complete picture of the model’s performance across different datasets. Figure 7a and Table 3 show the prediction results of ResNet50 on the sets of HSIs and ORIs. The superiority of ResNet50 in the HSI set was evident, where the accuracy reached 96% compared with 93% in the ORI set. Additionally, the Precision, Recall, Specificity, and f1-score in the HSI set were higher than those in the ORI set. Specifically, the Precision, Recall, Specificity, and f1-score in the HSIs set were 96%, 96%, 95%, and 96%, respectively, while the results in the ORI set were 95%, 95%, 92%, and 95%, respectively. In the case of Inception_v3 (Figure 7b), the Recall rate obtained in the HSIs was 85%, which is significantly higher than the rate achieved in the ORIs (75%). Consequently, the f1-scores, which reflect the harmonic mean of Precision, Recall, and Specificity, were also higher in the HSIs than in the ORIs, with values of 88% and 78%, respectively.

Given that the network architecture of GoogLeNet is similar to that of Inception_v3, their results were not significantly different. As shown in Figure 5c, the Recall rate obtained in the HSIs was 77%, which is 10% higher than the rate achieved in the ORIs (67%). Similarly, the F1-score was 7% higher in the HSIs than in the ORIs, with values of 87% and 80%, respectively. Figure 5d shows the prediction results of EfficientNet_B0. The findings demonstrate the superiority of EfficientNet_B0 in the HSI set, where the accuracy reached 97% compared with 94% in the ORIs set. Furthermore, the Precision, Recall, Specificity, and f1-score in the HSI set were higher than those in the ORI set. Specifically, the values were 99%, 92%, 99%, and 96%, respectively, in the HSI set and 94%, 91%, 91%, and 94%, respectively, in the ORI set.

Recall is an important indicator used to evaluate the performance of a model in detecting diseases. A higher recall rate indicates that fewer sick patients are identified as healthy, which can prevent the disease from worsening. The recall rate in the four models varied as follows: ResNet50: 95%, Inception_v3: 75%, GoogLeNet: 67%, and EfficientNet: 91%. In the case of HCQ hyperspectral image detection, the recall rates (Recall) were 96% for ResNet50, 85% for Inception_v3, 77% for GoogLeNet, and 92% for EfficientNet. These results indicate that the addition of hyperspectral images significantly improves the detection ability of HCQ. Through the comparison of different models, we can confirm that the use of HCQ color fundus images can be effectively predicted by deep learning. The accuracy of the HSI results reached more than 90% in the models tested, which can assist doctors in identifying HCQ as a tool. More HCQ color fundus images should be collected in the future to improve the accuracy of the results and promote their use for evaluation.

## 4. Discussions

### 4.1. Effects of Aging on Hydroxychloroquine Retinopathy

Hydroxychloroquine retinopathy is known to result in the destruction of rods and cones while leaving the cones relatively intact. This characteristic pattern often manifests as a bull’s-eye appearance. As a consequence of the damage to photoreceptors, the RPE may migrate into the regions affected, leading to the detection of pigment-filled cells within the outer nuclear layer and outer plexiform layer.

Age has been identified as a significant risk factor for HCQ retinopathy, as stated in a report by The American Academy of Ophthalmology [31]. Numerous studies have documented cases of HCQ retinopathy specifically occurring in elderly individuals. One study, in particular, demonstrated that electroretinography was capable of detecting changes in elderly patients (over 65 years of age) undergoing HCQ treatment, whereas such changes were not observed in younger patients [11]. This suggests that the likelihood of HCQ retinopathy incidence may increase with advancing age. It is plausible that HCQ induces abnormalities in the organization of the eye, leading to the destruction of rods and cones. Furthermore, it may affect structures with complex surface profiles, thereby contributing to the development of retinopathy associated with HCQ usage.

Figure 2c,d illustrate OCT images of patients affected by HCQ infection and normal patients. In all four eyes, OCT imaging revealed abnormalities in the external retina corresponding to areas identified as hydroxychloroquine-associated retinopathy through ophthalmoscopy results. These abnormalities involve the complete loss of the inner/outer segment junction of photoreceptors, while the RPE, outer limiting membrane, and translocation exhibit relative preservation. Furthermore, the surface profile of HCQ patients appears to be rough, exhibiting more folds compared to normal cases. This rough surface profile contributes to a scattering phenomenon when observed using hyperspectral imaging. Figure 5 demonstrates these differences, particularly in the long wavelength region, at five surveyed locations. The most pronounced difference is observed at the foveal location (F), as well as the superior (S1) and inferior (I1) locations. In our study, we selected three patients from three different age groups (60s, 70s, and 80s). This suggests that age can be considered a non-independent factor contributing to HCQ retinopathy.

### 4.2. The Relationship of HCQ Retinopathy and Diabetes Remains Uncertain

The relationship between HCQ retinopathy and diabetes has not been extensively investigated. A study examining the progression of diabetic retinopathy (DR) from mild to moderate and severe stages has suggested that HCQ may have an impact on the development of DR [32]. However, it is important to note that these findings are based on clinical trials with a limited sample size. In Figure 6, the spectral regions do not exhibit significant differences among the four stages of diabetes. This indicates that the stages of diabetes primarily affect the vascular system, including the arteries and veins within the eye. On the other hand, the structural abnormalities associated with HCQ retinopathy are relatively small and occur within the retina, often with minimal external manifestations.

### 4.3. A Novel Screening Technique for HCQ Retinopathy

To gain insights into the deep learning model and visually identify regions with prominent features, the Grad-CAM method [33] is employed to assess layer weights by generating heatmap distributions of feature layers. Gradient computation is performed at the final layer of the feature module in each deep learning model. Figure 8 illustrates the feature maps obtained for four deep learning models using HSIs of HCQ cases. The feature maps demonstrate that the heatmap scores are predominantly concentrated around the foveal locations (F). This observation aligns with previous studies on HCQ retinal retinopathy, which have utilized OCT spectral-domain analysis [34,35,36,37,38,39]. These studies have indicated that the concentration and presence of HCQ active ingredients significantly impact the thickness in the corresponding area. This provides a solid foundation for diagnosing HCQ retinopathy through screening using ophthalmoscope images and spectral analysis with OCT images. Moreover, the hyperspectral conversion algorithm enhances the detection of anomalies in the spectral-domain information, enabling computer-aided diagnosis models to accurately detect HCQ retinopathy.

## 5. Conclusions

This study analyzed the spectral differences caused by age changes in normal color fundus images. Different ages will cause spectral changes due to the atrophy of blood vessels in the fundus. The color fundus images of patients taking HCQ were analyzed and found to be significantly different in the wavelength of 500–600 nm, which could be due to retinopathy caused by HCQ. An additional study of diabetic color fundus images was conducted, and the result showed that diabetes did not cause changes in the spectrum.

Diseases related to the retina were previously thought to only occur due to aging. However, the age of onset is gradually decreasing, and continuous innovation can provide better care for detecting and treating patients. In this study, hyperspectral conversion technology was used to analyze the spectrum of color fundus images. The technology can effectively detect and recognize differences in the image spectrum after HCQ.

The development of AI in medical imaging has been ongoing for a long time, and various models have produced different prediction results for the same data. This study employed a hyperspectral conversion algorithm to analyze the spectrum of color fundus images and obtained HSIs to capture additional features. Using four deep learning models, namely, ResNet50, Inception_v3, GoogLeNet, and EfficientNet, we trained and evaluated HCQ ophthalmoscope images. HSI effectively improved the accuracy and other indicators in different models, indicating that hyperspectral conversion technology is beneficial for the analysis of color fundus images.

The application of retinal imaging and AI is widespread in the diagnosis of diseases such as glaucoma, DR, and other related conditions. However, each disease requires specialized treatment for specific lesions. Most research using hyperspectral imaging involves obtaining spectra by using spectrometers and analyzing the data to identify the presence or absence of lesions. In the future, hyperspectral retinal images could lead to the development of a system that can detect various eye diseases. Analyzing color fundus images for eye diseases can assist doctors in diagnosis and extend the deployment of telemedicine systems to bridge the gap between urban and rural areas, which often lack medical resources.

One limitation of our research is the underutilization of the spectral information inherent in hyperspectral images. Currently, we have only converted the hyperspectral data into images with three channels, thereby losing a significant amount of spectral detail. However, our objective is to explore advanced processing methods, specifically convolutional neural networks (CNNs), to extract and fully leverage the wealth of information present in hyperspectral data.

Our aim is to design a neural network architecture that is specifically tailored to effectively represent and exploit the rich spectral information provided by hyperspectral data. By developing such an architecture, we anticipate enhancing the effectiveness and efficiency of our analysis and obtaining more comprehensive insights from the hyperspectral datasets. This will allow us to maximize the potential of hyperspectral imaging in our research and unlock new opportunities for improved analysis and interpretation.

## Figures and Tables

**Figure 1 diagnostics-13-02373-f001:**
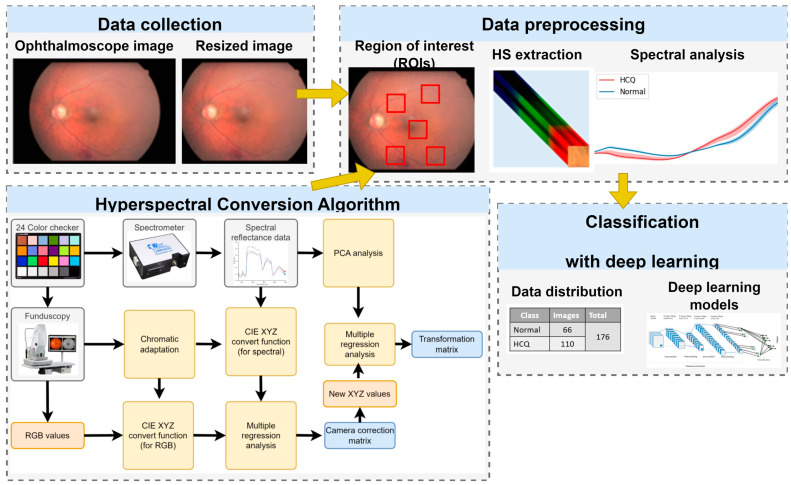
Experimental flow chart. Hyperspectral conversion algorithm was employed to extract spectral features from five positions as regions of interest (ROIs) in the augmented dataset. Spectral analysis was then applied to the ROIs to find the most prominent spectral regions. Band selection from range 500–600 nm is applied to perform color reproduction from hyperspectral band to RGB images. The HS images were transformed from HS cubes (512 × 512 × 401) to fit the input of deep learning models (512 × 512 × 3). Eventually, the HS images were utilized to train the deep learning models.

**Figure 2 diagnostics-13-02373-f002:**
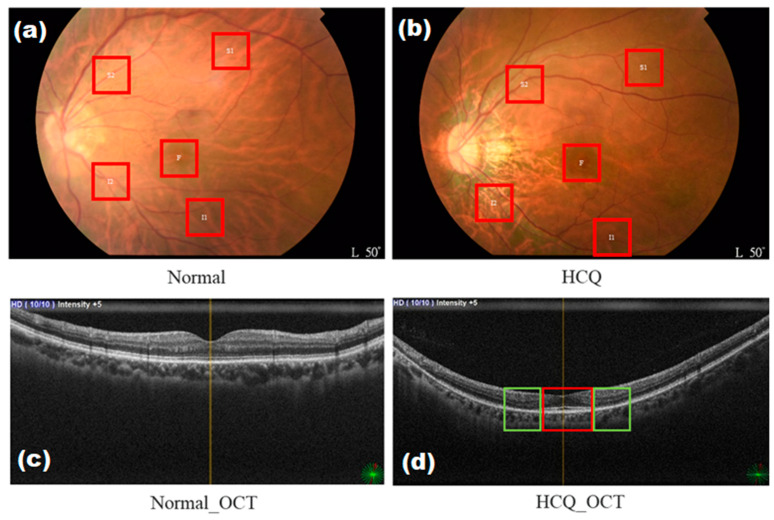
Color fundus images of HCQ including (**a**) normal, (**b**) HCQ, (**c**) normal OCT, and (**d**) HCQ OCT. The green box indicates thinning of the outer nuclear layer on both sides of the fovea, while the red box indicates a saucer-like appearance.

**Figure 3 diagnostics-13-02373-f003:**
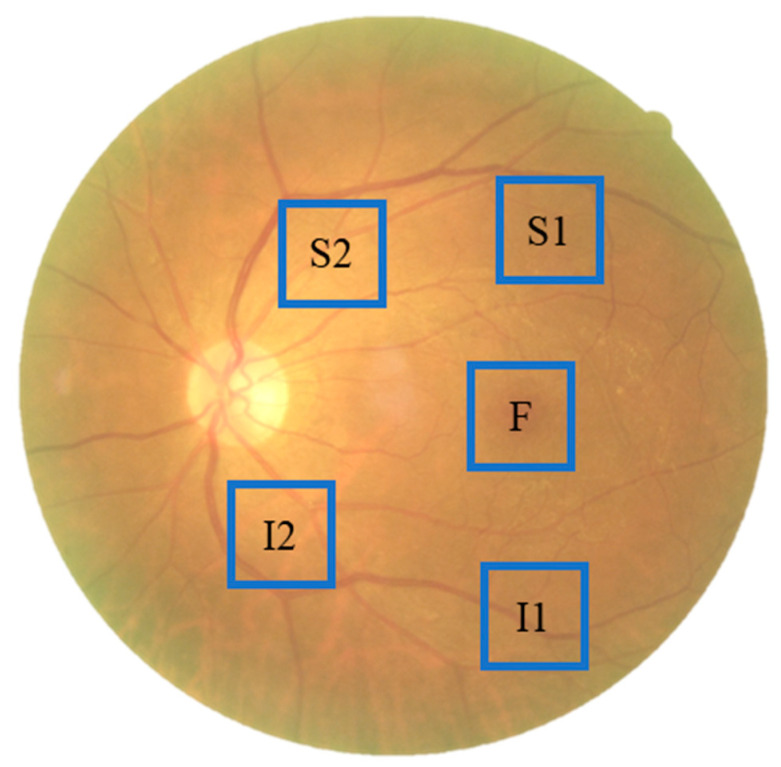
Schematic diagram of the cropped area of the fundus color image, where we set the sampling positions above the temporal vessel arcade (S1, S2), the fovea (F), and below the temporal vessel arcade (I1, I2).

**Figure 4 diagnostics-13-02373-f004:**
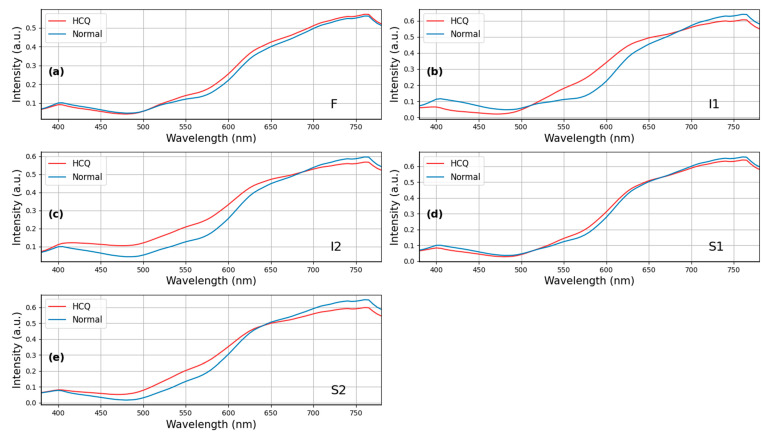
Retinal spectrum pairwise comparisons show differences between HCQ group (*n* = 25) and Normal group (*n* = 66) at sampling locations (**a**) F, (**b**) I1, (**c**) I2, (**d**) S1, and (**e**) S2. Specifically, positions I1, I2, S1 and S2 show the difference in the wavelength range from 500 to 650 nm. At position F, there is a spectral difference in the long wavelength region (>550 nm).

**Figure 5 diagnostics-13-02373-f005:**
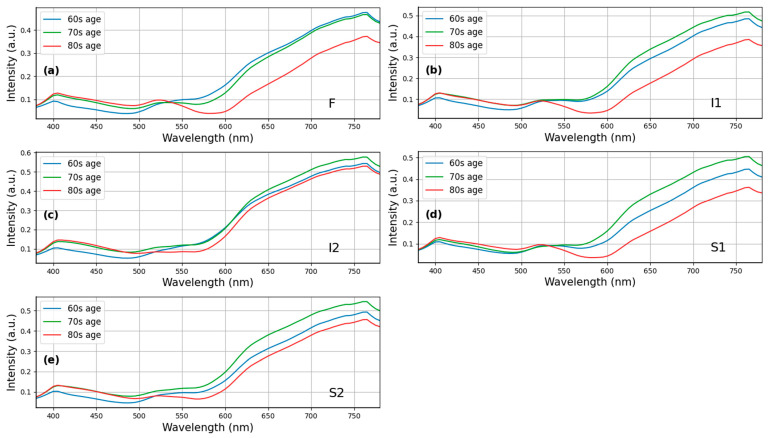
Comparison of normal retinal spectra by three age intervals (*n* = 91, *p*-value = 0.75, 95% CI: −5.51–4.86, two-tailed unpaired *t*-test) at sampling locations (**a**) F, (**b**) I1, (**c**) I2, (**d**) S1, and (**e**) S2. Positions F, I1 and S1 show the difference at long wavelengths (>550 nm).

**Figure 6 diagnostics-13-02373-f006:**
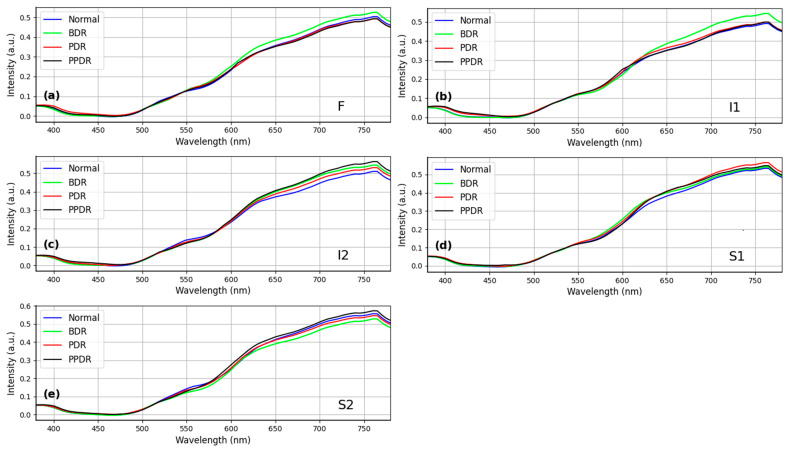
Comparison of Diabetic Retina Spectrum by stages between HCQ group at four levels of DR (*n* = 25, *p*-value = 0.56, chi-square test) at sampling locations (**a**) F, (**b**) I1, (**c**) I2, (**d**) S1, and (**e**) S2.

**Figure 7 diagnostics-13-02373-f007:**
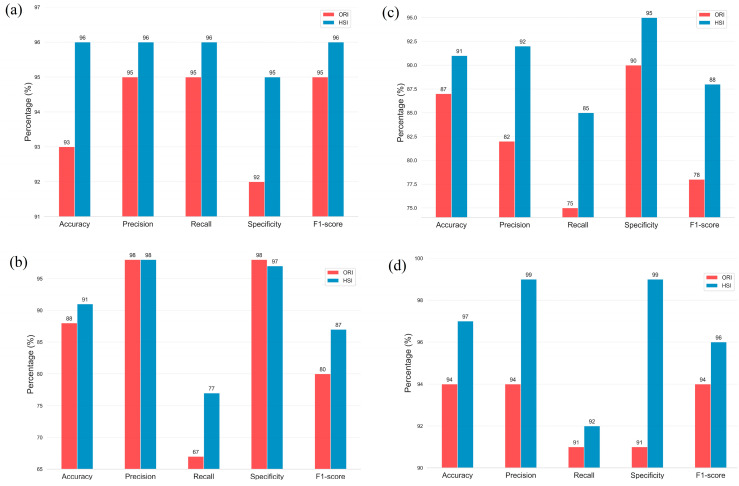
Comparison of Accuracy, Precision, Recall, Specificity, and F1-score among different models including (**a**) ResNet50, (**b**) Inception_v3, (**c**) GoogLeNet, and (**d**) EfficientNet_B0, in two datasets, namely, ORIs and HSIs.

**Figure 8 diagnostics-13-02373-f008:**
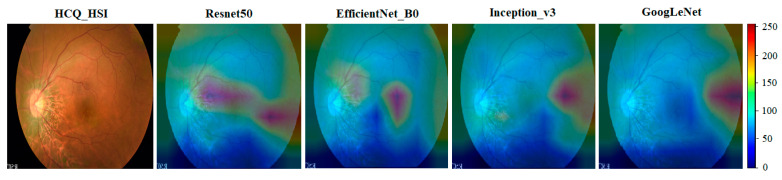
Visualization of feature maps. The visualization of feature maps in deep learning models reveals a prominent concentration of activations in the Resnet50 and EfficientNet_B0 architectures. The visualization of feature maps indicates that the heatmap distribution is predominantly concentrated in the foveal area. This concentration can effectively reveal the specific location of hydroxychloroquine (HCQ) formation, offering valuable insights for diagnosing HCQ retinopathy.

**Table 1 diagnostics-13-02373-t001:** Data statistics.

	HCQ Cases (*n* = 25)	Normal (*n* = 66)	*p*-Value	95% CI	Effect Size
Age ^1^	75.24 ± 8.41	75.83 ± 7.52	0.75	−5.51–4.86	−0.07
60 s age ^3^	*n* = 15	62.90 ± 3.57	<0.0001	60.69–65.11	−4.03
70 s age ^3^	*n* = 47	74.97 ± 2.27		74.23–75.72	−6.83
80 s age ^3^	*n* = 29	82.50 ± 1.92		81.61–83.39	−3.58
Sex (female/male) ^2^	22/3	45/21	0.10	−0.02–0.05	0.02
HBP (pos/neg) ^2^	17/8	59/7	0.03	−0.02–0.07	0.02
Glaucoma (pos/neg) ^2^	6/19	12/54	0.74	−0.003–0.011	0.003
AMD (pos/neg) ^2^	17/8	49/17	0.74	−0.004–0.011	0.004
DR (normal/BDR/PDR/PPDR) ^2^	6/6/4/9	21/11/18/16	0.43	0.54–3.58	0.17

HBP, high blood pressure; AMD, age-related macular degeneration; DR, diabetic retinopathy; BDR, background retinopathy; PDR, preproliferative retinopathy; PPDR, proliferative retinopathy, pos/neg positive/negative cases. ^1^ Interval variables are expressed as mean ± standard deviation. An unpaired two-tailed *t*-test is applied to analyze the mean of two independent HCQ and normal groups. The effect size and 95% CI are the difference between means. ^2^ Categorial variables are expressed as number of participants. The chi-square test evaluates whether there is a significant association between the categories of the two variables. The effect size and 95% CI are the odds ratio. ^3^ All participants in the 60s, 70s, and 80s age ranges were analyzed with a one-way analysis of variance (ANOVA) test.

**Table 2 diagnostics-13-02373-t002:** Data distribution.

	Train	Test	Total
Normal	53	13	66
HCQ	88	22	110
Total	141	35	176

**Table 3 diagnostics-13-02373-t003:** Results for the assessment criteria of HCQ.

		ORI	HSI
ResNet50	Accuracy	0.93	0.96
	Precision	0.96	0.96
	Recall	0.95	0.96
	Specificity	0.92	0.95
	F1-score	0.95	0.96
Inception_v3	Accuracy	0.87	0.91
	Precision	0.82	0.92
	Recall	0.75	0.85
	Specificity	0.90	0.95
	F1-score	0.78	0.88
GoogLeNet	Accuracy	0.88	0.91
	Precision	0.98	0.98
	Recall	0.67	0.77
	Specificity	0.98	0.97
	F1-score	0.80	0.87
EfficientNet_B0	Accuracy	0.94	0.97
	Precision	0.94	0.99
	Recall	0.91	0.92
	Specificity	0.91	0.99
	F1-score	0.94	0.96

## Data Availability

The data presented in this study are available in this article.

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
