# Peer review of "Detection of Hydroxychloroquine Retinopathy via Hyperspectral and Deep Learning through Ophthalmoscope Images"

_diagnostics, 2023, doi:10.3390/diagnostics13142373_

Round 1
Reviewer 1 Report
The manuscript by Wen-Shuang Fan et al. describes the use of deep learning based pseudo-spectral imaging for the detection of hydroxychloroquine retinopathy. It is an interesting approach dedicated to the development of digital pathology. It is included in the special issue of the Diagnostics Journal. The additional proofreading and error correction are necessary. However, the descriptions in the manuscript are sometimes confusing and it is difficult to understand directly how the research was carried out. Authors should really make an effort to adequately and clearly describe what was done and how it was done. My detailed comments are given below.
Major comments:
1) First of all, the title, the term "intelligent" is vague in my opinion. The authors should reconsider the change to "deep-learning based". Also, I have doubts that this technique should be called hyperspectral (see comments below).
2) The introduction is somewhat chaotic in my opinion. The authors should clearly state what the main aim of the research was, as it is rather vague in its current form.
3) From the description provided, it is difficult to understand whether hyperspectral spectra of the retinal areas were recorded here directly or were indirectly extracted/converted from the recorded images of 24 colour cards. It is really confusing. In my opinion, this technique should not be called hyperspectral imaging at all. More like pseudo-hyperspectral. Moreover, the hyperspectral imaging technique is characterised by high spectral resolution, but it was not indicated in the manuscript.
4) The camera of the ophthalmoscope was not equipped with a NIR filter? What is the sensitivity of the used detector in ophthalmoscope (Nikon Digital SLR Camera) as I found for Kowa Nonmyd 7.
5) In general, a description of the proposed measurement systems used for retinal characterisation is omitted here by the authors. There is a bit more detail in the supplementary material, but still important information is missing. In my opinion, the authors should correct this, as the whole concept of the research they present is based on the use of new phenotypic patterns of the retina in the form of hyperspectral spectra. The technical specifications of the OCT system used for the recording of the images are not provided. Authors are asked to provide the information about: what was the spectral range and central wavelength of the radiation source in the system used? what was the A-scan/B-scan rate during the OCT examination? what was the FOV of the OCT system used? what was the imaging depth of the OCT system used?
6) The technical specification of the used ophthalmoscope was also not provided by the authors. Authors used conventional colour fundus images registered for white light or for laser scanning ophthalmoscope images for specific wavelength. What was the spectral range of the used light source? What was the spectral resolution of the OceanOptics spectrometer?
7) How have the hyperspectral spectra been normalised to the spectral spectrum of the light source used and the sensitivity/exposure time of the detector used? The parameters of these two elements can significantly affect the hyperspectral data.
8) Taking into account the uncertainty in the determination of the hyperspectral spectrum, are the differences in the spectra related to the age of the patients, described in the Results section, statistically significant?
9) Fig.4: The spectra shown are averages of how many patients? This information should be provided.
Minor comments:
1) The quality/resolution of Fig.1 should be increased, it is difficult to read the text in the lower left diagram.
2) Line 187-189 “In the five regions of Figure 3, no significant difference in the spectrum was observed with diabetes, indicating that the HCQ fundus color image spectra did not vary with or without diabetes (Supplementary Information Section 2).” – Fig.3 not Fig. S3 or fig.4?
3) The sub-images in fig.S3 are not described
4) The quality/resolution of Fig.1 should be increased, it is difficult to read the text in the lower left diagram.
5) There are many repetitions in the manuscript, e.g. “Hyperspectral imaging conversion technology can be utilized to convert images into spectra and vice versa.” is in lines 84-85 and is repeated in line 141-142. Please correct such errors in all manuscript.
6) There are editing errors (e.g. line 104 “The The overall” ) in the manuscript which should be corrected before the manuscript is sent for review.
7) Line 42: „ A multitude of ocular diseases” -> rather "A wide range of " eye/ ocular diseases, "A variety of " eye/ ocular diseases
My comments about the quality of English language are listed directly in "Comments and Suggestions for Authors
".
Reviewer 2 Report
The paper is very interesting and of novelty. The authors present a new model based on AI that can help diagnosis of hydroxicloroquine induced retinopathy. The methods are clearly described .
The results are very interesting and promising.
However, I recommed that the Discussions should be enlarged and presented as a separated section. The limits of the study should be described
Reviewer 3 Report
The author used spectral analysis to discern differences between normal and hydroxychloroquine lesions on fundus images, which may be of clinical importance. Here are my comments in methods section:
1. Why did the author choose to try "ResNet50, Inception_V3, GoogLeNet and EfficientNet"? In other words, they are all deep learning neural networks, and there are other excellent neural networks (such as DenseNet), why did the author only try the ones mentioned in the current paper?
2. In addition to #1, the author should provide more information regarding the training of these models. For example, the author tried EfficientNet, how many layers did they used to training (EfficientNet V1 to V5?), how many epochs during training? What activation function did they use to train these models? These details would make their study more reliable and reproducible.
3. Please supply relative information of model training platform.
4. There were 110 images from 25 patients. How many images were taken from each patient? Were all the images from both eyes, or just from one eye? Because the author used 22 images for model testing, this information was important in void data leakage. How did the author divide the data into training and test data? As the author did not mention the validation data set, if the author used cross-validation, please add relative information?
5. Since the current study only included 25 positive patients, without an external dataset, how did the author evaluate the generalization ability of the currently trained models, rather than over-fitting?
6. As the author mentioned spectral reflectance differences between ages, please provide statistical methods as well as actual images used for data calculations in the methods section.
7. Deep leaning models were born with overfitting. The current study did not test the generalization ability of the models, so it's hard to say that the models could be useful in general. If the author was only trying to prove the usefulness of hyperspectral images, machine learning would be quick and easy to be handled. Otherwise, if both simple machine learning models, such as supported vector machines, and deep learning models were tried, it would be more convincing for the current study.
Round 2
Reviewer 1 Report
I thank the authors for the corrections and clarifications made. However, it seems to me that some of my previous comments are still valid:
1) Do the authors consider the statistical significance analysis of the effect of patient age on hydroxychloroquine retinopathy sufficient? The presented results of the statistical analysis refer to 3 patients, which is far too small sample size. Which test was used to determine the p-value?
2) Furthermore, section 4.1 should be entitled "Effect of ageing on hydroxychloroquine retinopathy" and not " Effects of hydroxychloroquine retinopathy on aging". Because retinopathy is unlikely to affect the ageing process.
3) The presented comparison of the size of the learning and training set for each of the 3 classes shows that both sets are significantly less numerous than the learning set. After all, this can affect the classification results. The Authors did not include any analysis or discussion about this issue.
4) Why didn't the authors decide to increase the number of the patterns/images e.g. by modifying their rotation, reflection, shift etc.? This can be done in a very simple way.
5)The main advantage of hyperspectral imaging is the ability to analyse spatially-resolved information about the absorption properties of objects. Therefore, one wonders why the authors chose to analyse only the spectral spectra in detail. The analysis of the spatial information is limited to Fig.8 only.
6) Fig.8: Can Authors add the colorbar of the used heatmap.
In my opinion, the manuscript still needs to be revised.
Reviewer 2 Report
The authors revised the manuscript according to the recommendations. I have no further issues.
Author Response
Thanks for the comments. We really appreciate the comments of reviewers.
Reviewer 3 Report
All responses are reasonable and acceptable.
Author Response

(The authors gave the same response as above.)

Round 3
Reviewer 1 Report
I would like to thank the authors for the clarifications they provided and the changes they made to the manuscript. The inclusion of more data always leads to a more meaningful analysis of the results and the scientific quality of the publication. Congratulations to the authors. In my opinion, the current version of the manuscript is suitable for publication.
Reviewer 2 Report
I recommend accept. The manuscript has been revised.
Author Response

(The authors gave the same response as above.)
